# Meta-analysis of dispensable essential genes and their interactions with bypass suppressors

Carles Pons[1] , Jolanda van Leeuwen[2]

**Genes have been historically classified as essential or non-essential based on their requirement for viability. However, genomic mutations can sometimes bypass the requirement for an essential gene, challenging the binary classification of gene essentiality. Such dispensable essential genes represent a valuable model for understanding the incomplete penetrance of loss-of-function mutations often observed in natural populations. Here, we compiled data from multiple studies on essential gene dispensability in *Saccharomyces cerevisiae* to comprehensively characterize these genes. In analyses spanning different evolutionary timescales, dispensable essential genes exhibited distinct phylogenetic properties compared with other essential and non-essential genes. Integration of interactions with suppressor genes that can bypass the gene essentiality revealed the high functional modularity of the bypass suppression network. Furthermore, dispensable essential and bypass suppressor gene pairs reflected simultaneous changes in the mutational landscape of *S. cerevisiae* strains. Importantly, species in which dispensable essential genes were non-essential tended to carry bypass suppressor mutations in their genomes. Overall, our study offers a comprehensive view of dispensable essential genes and illustrates how their interactions with bypass suppressors reflect evolutionary outcomes.**

## Introduction

Identification of the genes required for viability is key for both fundamental and applied biological research. Essential genes constrain genome evolution (Jordan et al, 2002; Bergmiller et al, 2012; Luo et al, 2015), define core cellular processes (Wang et al, 2015), identify potential drug targets in pathogens and tumors (Roemer et al, 2003; Behan et al, 2019), and are the starting point to determine minimal genomes (Juhas et al, 2011; Hutchison et al, 2016). The fraction of essential genes within a genome reflects its complexity and redundancy and anticorrelates with the number of encoded genes (Rancati et al, 2018). For instance, 80% of 482 genes in *Mycoplasma genitalium* (Glass et al, 2006), 18% of ~6,000 genes in *S. cerevisiae* (Giaever et al, 2002), and only 10% of the ~20,000 genes in human cell lines (Blomen et al, 2015; Hart et al, 2015; Wang et al, 2015) are essential for viability. Essential genes tend to code for protein complex members (Dezso et al, 2003; Hart et al, 2007), play central roles in genetic networks (Costanzo et al, 2010), have few duplicates (Giaever et al, 2002), and share other properties (Deng et al, 2011; Hart et al, 2015) that differentiate them from non-essential genes, enabling their prediction (Hwang et al, 2009; Lloyd et al, 2015; Zhang et al, 2016). Although gene essentiality is significantly conserved, essentiality changes are frequent across species and even between individuals. For instance, 17% of the 1:1 orthologs between *S. cerevisiae* and *Schizosaccharomyces pombe* have different essentialities (Kim et al, 2010). Also, 57 genes differ in essentiality between two closely related *S. cerevisiae* strains (Dowell et al, 2010), and a systematic analysis of 324 cancer cell lines from 30 cancer types found that only ~40% of essential genes were shared across cell lines (Behan et al, 2019). Thus, essentiality is not a static property, and changes in the genetic background can change the essentiality of a gene (Rancati et al, 2018).

Recently, we and others have systematically identified essential genes that are non-essential (i.e., dispensable essential genes [DEGs]) in the presence of suppressor mutations (i.e., the genetic changes enabling the bypass of gene essentiality) in *S. cerevisiae* (Liu et al, 2015; van Leeuwen et al, 2020) and *S. pombe* (Li et al, 2019b; Takeda et al, 2019). Both DEGs and their bypass suppressors exhibit specific features that differentiate them from other essential genes (i.e., core essential genes) and passenger mutations (i.e., randomly acquired mutations without an effect on fitness). For instance, DEGs are more likely to have paralogs, to be absent in other species, and to encode members of smaller protein complexes compared with core essential genes (Liu et al, 2015; van Leeuwen et al, 2020), whereas suppressor genes tend to be functionally related to the DEG (van Leeuwen et al, 2020). We previously exploited the specific properties of these genes for their accurate prediction (van Leeuwen et al, 2020).

Identification of the suppressor genes responsible for bypassing the requirement for the essential gene is important to dissect the function of both genes (van Leeuwen et al, 2016), to expose the genetic architecture of phenotypic traits (Mackay, 2014; Wei et al, 2014), and to

---

[1]Institute for Research in Biomedicine (IRB Barcelona), The Barcelona Institute for Science and Technology, Barcelona, Spain   [2]Center for Integrative Genomics, Bâtiment Génopode, University of Lausanne, Lausanne, Switzerland

Correspondence: carles.pons@irbbarcelona.org

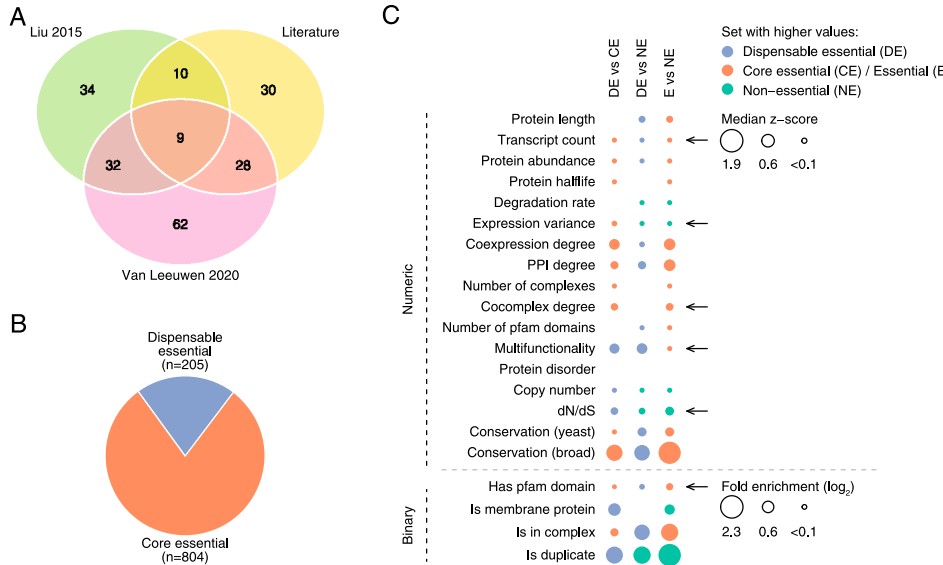

**Figure 1. Properties of dispensable essential gene sets.**
**(A)** Number of dispensable essential genes per individual dataset and their overlap. **(B)** Fraction of dispensable and core essential genes in the combined dataset. Labels include the number of genes in each category. **(C)** Enrichment of dispensable essential versus core essential genes (left column), dispensable essential versus non-essential genes (center column), and essential versus non-essential genes (right column) for a panel of 21 gene features. Top and bottom panels include numeric and binary features, respectively. Dot size is proportional to the median z-score and fold enrichment, respectively, and only enrichments with *P* < 0.05 (Mann–Whitney *U* test and Fisher's exact test, respectively) and FDR < 10% are shown. The arrows indicate properties of dispensable essential genes not previously identified.

understand drug resistance mechanisms (Woodford & Ellington, 2007). Suppressor mutations could also explain the presence of presumably highly detrimental genetic variants in natural populations (Jordan et al, 2015; Narasimhan et al, 2015; Chen et al, 2016). For instance, highly penetrant disease-associated mutations are sometimes present in healthy individuals (Chen et al, 2016), and human pathogenic variants can be fixed in other mammalian species without obvious deleterious consequences (Jordan et al, 2015). However, whether suppression interactions identified in laboratory strains are relevant in natural evolutionary landscapes and could explain the presence of deleterious genetic variants in populations remains an open question.

Here, we compiled a comprehensive set of DEGs in *S. cerevisiae* identified across different studies to exhaustively compare their properties to core essential and non-essential genes, with a particular focus on phylogenetic features. We integrated bypass suppressor genes into an interaction network with DEGs to identify prevalent interaction motifs and to analyze the relationship of bypass suppression pairs in other species. This work presents a systematic characterization of DEGs and explores how their interactions with suppressors reflect evolution in natural populations.

## Results

### Dispensable essential gene datasets

We compiled a comprehensive list of DEGs in *S. cerevisiae* from two large-scale studies (Liu et al, 2015; van Leeuwen et al, 2020) and from individual cases described in the literature (van Leeuwen et al, 2020) (Fig 1A). We only considered studies in which gene essentiality was bypassed in a laboratory yeast strain, as these often involve a single causal bypass suppressor gene, and disregarded studies that focused on essentiality changes across natural yeast strains, which are frequently driven by complex combinations of genetic variants (Dowell et al, 2010; Chen et al, 2022; Wang et al, 2022). In total, 205 DEGs

had been identified, representing ~20% of all tested essential genes (Fig 1B). Cases of bypass suppression were identified by looking for rare survivors in populations of 100–150 million cells deleted for an essential gene (van Leeuwen et al, 2020; experimental dataset), by following germination of single deletion mutant spores (Liu et al, 2015), or by a mixture of methods, including directly testing the effect of a mutation on essential gene deletion mutant viability (van Leeuwen et al, 2020; literature dataset). These methodological differences could possibly affect the detected DEGs.

To determine whether the datasets could be merged, we compared various properties of the DEGs described in each dataset. The DEGs identified in the three datasets overlapped significantly (*P* < 0.001, randomization test). In all datasets, DEGs showed similar functional enrichments (Fig S1A) and were depleted for fundamental cellular processes like RNA processing or translation and enriched for more peripheral functions related to signaling or transport (*P* < 0.05, Fisher's exact tests, and false discovery rate [FDR] < 10%). Furthermore, protein complexes tended to be either completely dispensable or indispensable across datasets (*P* < 0.05 in the combined dataset, randomization test, Fig S1B). For instance, the combined dataset contained 14 protein complexes with only dispensable essential subunits (Fig S1C), significantly more than expected by chance (*P* < 0.002, randomization test, Fig S1B). DEGs were more likely than core essential genes to be non-essential in the closely related *S. cerevisiae* strain Sigma1278b (*P* < 0.0005 in the combined dataset, Fisher's exact test, Fig S1D), and to be absent in the *S. cerevisiae* core pangenome (*P* < 0.05 in the combined dataset, Fisher's exact tests, Fig S1E). Because the properties of the combined and individual datasets were similar, we used the combined dataset in the following analyses.

### Properties of dispensable essential genes

By querying an extensive panel of 21 gene features (see the Materials and Methods section, Fig S1F), we compared the properties of

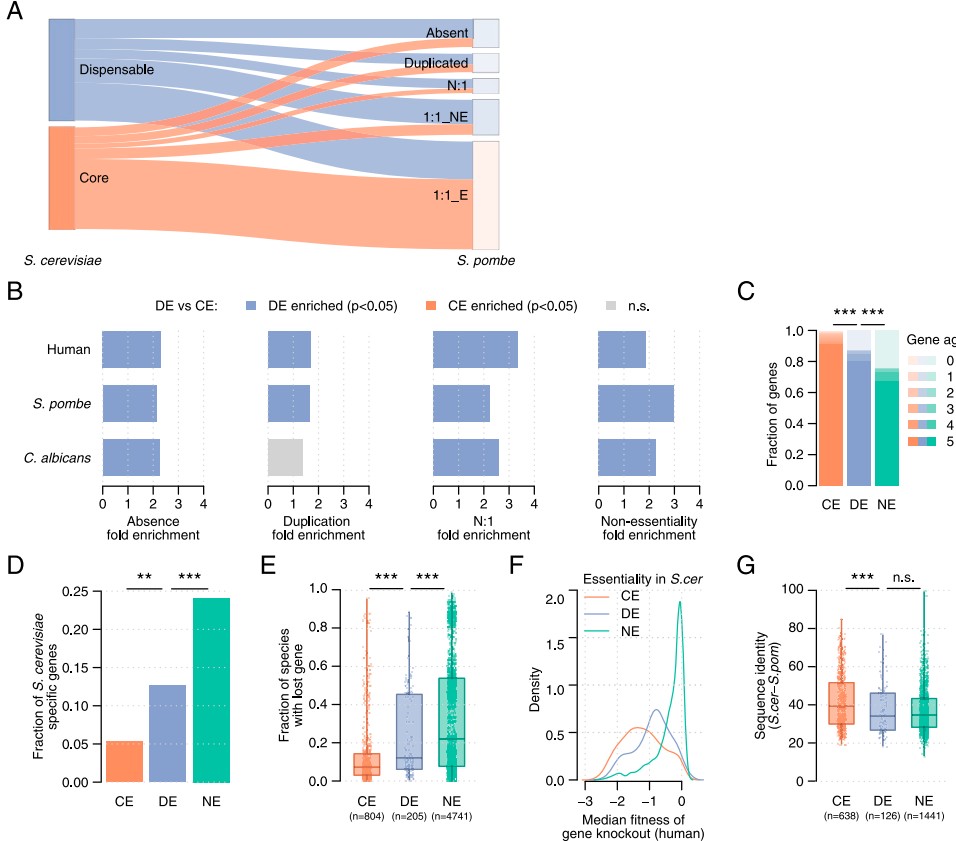

Figure 2. Phylogenetic analysis of dispensable essential genes.

**(A)** Orthology relationships in *S. pombe* of dispensable and core essential *S. cerevisiae* genes. The fraction of absent, duplicated, N:1, and essential and non-essential 1:1 orthologs is shown for each gene set. **(B)** Fold enrichment of dispensable essential *S. cerevisiae* genes with respect to core essential genes for absence, duplication, N:1 relationships, and non-essential 1:1 orthologs in *S. pombe*, *C. albicans*, and human. Purple and orange bars identify significant enrichments (*P* < 0.05, Fisher's exact test) with higher overlaps for dispensable essential and core essential genes, respectively (see Table S2 for details). Grey bars identify non-significant enrichments. **(C)** Fraction of genes within each age group, ranging from zero (found only in *S. cerevisiae*) to five (found in the furthest ancestor), for the three sets of genes. **(D)** Fraction of genes with age zero (*S. cerevisiae* specific) for each gene set. **(E)** Fraction of gene loss events across species for each *S. cerevisiae* gene grouped by gene set. **(F)** Median fitness per gene knockout across a panel of cancer cell lines. Genes are grouped by their essentiality in *S. cerevisiae*, and the density is shown. **(G)** Protein sequence identity between gene products in *S. cerevisiae* and 1:1 orthologs in *S. pombe*. **(C, D, E, F, G)** CE, core essential; DE, dispensable essential; NE, non-essential. Statistical significance was calculated using Fisher's exact (D) and Mann–Whitney *U* tests (C, E, F, G). n.s., not significant; *P < 0.05; **P < 0.005; ***P < 0.0005.

dispensable and core essential genes and found several significant differences (*P* < 0.05 using Mann–Whitney *U* tests for numeric features and Fisher's exact tests for binary features, and FDR < 10%). DEGs tended to exhibit more stable gene expression levels and lower transcript counts, to be less conserved across species, to have more gene duplicates and higher evolutionary rates, and to be coexpressed with fewer genes than core essential genes. The proteins encoded by DEGs tended to be more multifunctional, to lack structural domains, to localize to a membrane, to be absent from protein complexes, and to have fewer protein–protein interactions, lower abundances, and shorter half-lives compared with those encoded by core essential genes (Fig 1C and Table S1). Interestingly, the observed differences between dispensable and core essential genes resembled the differences between non-essential and essential genes (Fig 1C and Table S1). Thus, we asked whether dispensable essential and non-essential genes shared the same properties and found that they comprised two different classes of genes with clearly distinct features (Fig 1C and Table S1). Broadly, features of DEGs fell between those of core essential and non-essential genes, consistent with and extending previous findings in a smaller dataset (Liu et al, 2015).

## Phylogenetic analysis of dispensable essential genes

We further explored the differences in gene conservation between dispensable and core essential genes using the phylogeny of *S. cerevisiae*, starting with a large panel of sequenced *S. cerevisiae* strains (Peter et al, 2018). DEGs were more likely than core essential genes, but less than non-essential genes, to harbor deleterious mutations disrupting protein sequences (*P* < 0.0005, Fisher's exact test, Fig S2A), to present higher non-synonymous mutation rates (*P* < 0.0005, Mann–Whitney *U* test, Fig S2B), and to show copy number loss (CNL) events in other *S. cerevisiae* strains (*P* < 0.0005, Fisher's exact test, Fig S2C). To further investigate differences in the evolutionary pressure on dispensable essential and core essential genes, we analyzed essentiality data and orthology relationships in *Candida albicans*, *S. pombe*, and human cell lines (Figs 2A and S2D and E and Table S2). Genes that were dispensable essential in *S. cerevisiae* were more often absent than core essential genes in each of the analyzed species (*P* < 0.0005, Fisher's exact tests, Fig 2B). We hypothesized that this bias could be caused by: (i) genes specific to the *S. cerevisiae* phylogenetic branch and, thus, not present in their common ancestor or (ii) genes present in their common ancestor but lost in the phylogenetic branch of the analyzed species. To determine the contribution of each factor, we calculated the age of each *S. cerevisiae* gene by identifying the furthest species with an orthologous gene. DEGs were enriched for younger genes with respect to core essential genes (*P* < 0.0005, Mann–Whitney *U* test, Fig 2C), particularly for genes with no ortholog in any other species (i.e., specific to *S. cerevisiae*; *P* < 0.005, Fisher's exact test, Fig 2D). Next, for each species, we defined lost genes as those absent in that species but present in its common ancestor with

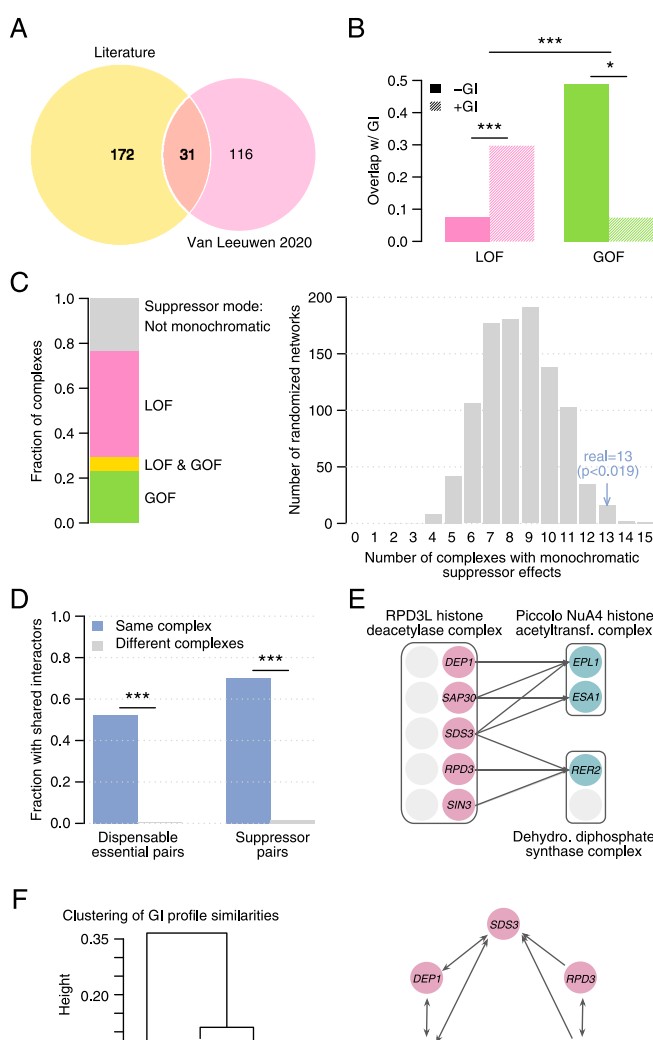

**Figure 3. Bypass suppression interaction network.**
**(A)** Number of bypass suppression gene pairs in each individual dataset and their overlap. **(B)** Fraction of loss-of-function (LOF) and gain-of-function (GOF) bypass suppression pairs that overlap with negative and positive genetic interactions. **(C)** (left) Fraction of monochromatic complexes in which all dispensable essential genes are suppressed by either LOF or GOF bypass suppressors. Only complexes with two or more dispensable essential subunits are shown. In one complex, all subunits could be suppressed by LOF suppressors but also by GOF suppressors (indicated by "LOF & GOF" in the panel). (right) Number of monochromatic complexes in the suppression bypass network (blue) and in 1,000 randomized networks (grey). **(D)** Fraction of gene pairs encoding members of the same complex and of different complexes that share an interactor. Dispensable essential gene pairs are shown on the left, bypass suppressor gene pairs on the right. **(E)** Interaction modularity of the bypass suppressor genes coding for members of the RPD3L histone deacetylase complex (CPX-1852). **(F)** Genetic interaction profiles of the bypass suppressor genes in (E). (left) Hierarchical clustering of the genetic interaction profiles; (right) network showing genetic interaction profile similarities above 0.2. **(B, D)** Statistical significance was calculated using Fisher's exact test. *$P < 0.05$; ***$P < 0.0005$.

*S. cerevisiae*. We found DEGs were more often lost in other species than core essential genes ($P < 0.0005$, Mann–Whitney $U$ test, Fig 2E). Thus, the absence of DEGs in other species can be explained both

by genes specific to *S. cerevisiae* and by gene loss events in those species.

Furthermore, DEGs present in other species were more frequently duplicated and had more N:1 orthology relationships ($P < 0.05$, Fisher's exact test, Fig 2B) than core essential genes. For genes with a 1:1 ortholog in other species, DEG orthologs were more often non-essential than orthologs of core essential genes ($P < 0.0005$, Fisher's exact test, Fig 2B), also in the closely related *Saccharomyces uvarum* species ($P < 0.05$, Fisher's exact test, Fig S2F). Similarly, fitness data from a panel of 1,070 cancer cell lines (Meyers et al, 2017) revealed that knockout of DEG orthologs led to less severe proliferation defects than knockout of core essential gene orthologs ($P < 0.0005$, Mann–Whitney $U$ test, Fig 2F). Thus, genes that can be bypassed by genetic mutations in *S. cerevisiae* tend to be non-essential in other species. We show the comparison between essential and non-essential genes and dispensable essential and non-essential genes to contextualize the observed differences (Fig S2G–I and Table S2).

Finally, we compared sequences of *S. cerevisiae* proteins and their 1:1 orthologs in *S. pombe* and *C. albicans*. Gene products of DEGs had lower sequence identity and differed more in sequence length than core essential proteins ($P < 0.05$, Mann–Whitney $U$ tests, Figs 2G and S2J–L), in line with the dN/dS data (Figs 1C and S2B). Overall, orthology relationships, phenotypic changes, and sequence divergence reflect that the evolutionary pressure on DEGs is more lenient than on core essential genes but more strict than on non-essential genes.

### The bypass suppressor interaction network

Identification of the relevant genetic changes (i.e., suppressors) required to tolerate the deletion of an essential gene is key to interpreting the presence of deleterious genetic variants in natural populations. To improve our knowledge on the mechanisms of genetic suppression, we built an interaction network between DEGs and their bypass suppressors by combining data from our recent systematic study (van Leeuwen et al, 2020) and the literature (van Leeuwen et al, 2020). The two individual suppression interaction networks overlapped significantly ($P < 0.001$, randomization test, Fig 3A) and were similarly enriched in functional associations ($P < 0.0005$, Fisher's exact tests, Fig S3A). The combined network included a total of 319 unique bypass suppression gene pairs, corresponding to 243 suppressors and 137 DEGs out of the 205 known DEGs. For the remaining DEGs (33% of the dataset), the suppressor variants have not been identified. Dispensable essential and suppressor genes tended to be functionally related ($P < 0.05$, randomization test, and FDR < 10%, Fig S3B), particularly for close functional relationships like cocomplex or copathway membership ($P < 0.0005$, Fisher's exact tests, Fig S3A), and suppressors related to nuclear-cytoplasmic transport and transcription processes were more frequent than expected by chance ($P < 0.05$, Fisher's exact test, and FDR < 10%, Fig S3B). For a subset of bypass suppressors, we and others have previously determined experimentally whether a suppressor mutation had a loss-of-function (LOF) or gain-of-function (GOF) effect, by testing the effect of suppressor gene mutation or overexpression on the viability of the corresponding DEG deletion mutant (van Leeuwen et al, 2020). Here, we found that

for 50% and 26% of the dispensable genes, only LOF and GOF suppressors had been identified, respectively, and in 15% of the cases, both types of suppressors had been described (Fig S3C). For the remaining cases, the nature of the suppressor had not been determined.

Genetic interactions identify combinations of mutants that result in unexpected phenotypes given the phenotypes of the individual mutants. In negative genetic interactions, the resulting phenotype is more severe than expected, whereas in positive genetic interactions, the phenotype is healthier than expected. In a bypass suppression interaction, a secondary mutation recovers the lethal phenotype caused by an essential gene deletion, therefore representing an extreme form of positive genetic interaction. We grouped interacting pairs in the bypass suppression network by their suppression mode (i.e., LOF or GOF) and evaluated their overlap with a global genetic interaction network (Costanzo et al, 2016), generated using hypomorphic alleles of essential genes and deletion alleles of non-essential genes (see the Materials and Methods section). We first analyzed bypass suppression gene pairs with LOF suppressors and found that LOF alleles of these gene pairs often had a positive genetic interaction with each other in the global network ($P < 0.0005$, Fisher's exact test, Fig 3B). In spite of the different experimental protocols, this overlap is expected because both bypass suppression and genetic interactions were identified using LOF alleles. Conversely, when analyzing bypass suppression gene pairs with GOF suppressors, we found that the corresponding LOF alleles mainly showed negative genetic interactions ($P < 0.05$, Fisher's exact test, Fig 3B). Thus, GOF and LOF alleles of the suppressor gene have opposite effects when combined with a LOF allele of the corresponding DEG, being beneficial or detrimental as shown by the bypass suppression and genetic interaction networks, respectively.

## Structure of the bypass suppression interaction network

Interaction density (i.e., the percentage of gene pairs with an interaction) of the bypass suppression network ranged from 0.007% to 0.96% depending on whether we considered all possible gene pairs or only pairs between the identified dispensable essential and suppressor genes, respectively. In spite of the sparsity of this network, several patterns emerge showing its structure and modularity. For instance, all DEGs in the same protein complex tended to interact with either GOF or LOF suppressors. These monochromatic interactions affected 13 out of 17 non-redundant protein complexes with at least two dispensable essential subunits in our dataset ($P < 0.05$, randomization test, Fig 3C), suggesting similar suppression types apply for functionally related genes. Importantly, both individual suppression networks contributed to this result (Fig S3D), discarding the potential bias from specific hypothesis-driven experiments in the literature dataset. We analyzed the topology of the network and found that for 45% of the DEGs, multiple suppressors had been described (Fig S3E). This set of genes exhibited specific features compared with DEGs for which only a single suppressor had been described (Fig S3F). For instance, DEGs with multiple identified suppressors tended to have higher multifunctionality and an increased number of structural domains ($P < 0.05$, Fisher's exact test, and FDR < 10%), which suggest multiple

different molecular mechanisms of suppression may exist for these DEGs. Suppressors were more specific than DEGs, and only 23% of them interacted with multiple genes (Fig S3E). Next, we explored the relationship between functional similarity and connectivity patterns. We found that genes in the same protein complex tended to have the same interactors: 52% of the DEGs encoding members of the same complex shared suppressor genes, and 70% of the suppressor genes encoding members of the same complex shared DEGs (Fig 3D), more than expected by chance ($P < 0.0005$, Fisher's exact test).

To illustrate the underlying modular structure of the bypass suppression interaction network, we explored the connectivity of *NCB2* and *BUR6*, both DEGs with known suppressors and the only two members of the negative cofactor 2 transcription regulator complex (ID CPX-1662 in the Complex Portal [Meldal et al, 2021]). *NCB2* and *BUR6* have seven and 10 identified bypass suppressor genes, respectively, six of which are in common, again showing that functionally related DEGs tend to share suppressors (Fig S3G). Two of these common suppressors belong to the core Mediator complex that plays a role in the regulation of transcription (CPX-3226), showing that interactors of the same dispensable gene tend to be functionally related both to each other and to the DEG they are suppressing. The other four shared suppressor genes also affect transcription and encode subunits of the transcription factor TFIIA complex (CPX-1633), the general transcription factor complex TFIIH (CPX-1659), and the DNA-directed RNA polymerase II complex (CPX-2662). Interestingly, the *NCB2*-specific suppressor, *TOA2*, also encodes a member of TFIIA, and three of the four *BUR6*-specific suppressors members of RNA pol II or Mediator, further illustrating the modularity of the network. In another example (Fig 3E), members of the RPD3L histone deacetylase complex (CPX-1852) suppress two different protein complexes. *DEP1*, *SAP30*, and *SDS3* suppress the two essential subunits of piccolo NuA4 histone acetyltransferase complex (CPX-3185), whereas *RPD3*, *SIN3*, and *SDS3* interact with the Rer2 subunit of the dehydrodolichyl diphosphate synthase complex (CPX-162). This modularity in the suppression interaction pattern of RPD3L subunits is also observed in genome-wide genetic interaction patterns, which are more similar for RPD3L subunits that suppress the same query gene than for RPD3L subunits that suppress functionally diverse query genes (Fig 3F). These patterns suggest a functional modularity within the complex which is supported by its modeled structure (Sardiu et al, 2009).

## Mutational landscape of *S. cerevisiae* strains reflects bypass suppression relationships

We wondered if the genetic dependencies described in the suppression interaction network were reflected in the genomic variation present in natural populations. Because homozygous deletions of essential genes are extremely rare across *S. cerevisiae* strains (median of one per strain), we first focused on DEGs with CNL events. Hemizygosity is associated with a decrease in gene expression levels and can impact cell growth (Pavelka et al, 2010), particularly in essential genes. For instance, even if only ~10% of essential genes were haploinsufficient under rich media conditions (Deutschbauer et al, 2005), this increased to 30–50% when more conditions and

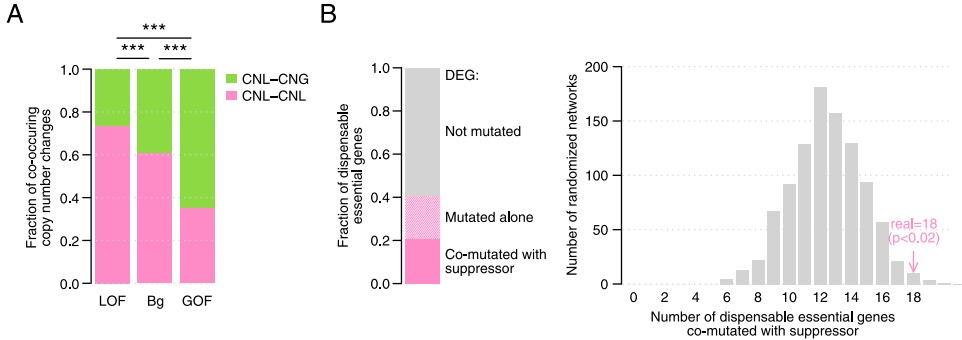

**Figure 4. Co-occurring mutations in *S. cerevisiae* strains.**
**(A)** Proportion of copy number co-loss and loss-gain (DEG–suppressor) events across a panel of *S. cerevisiae* strains for bypass suppression gene pairs in which the suppressor carried either a LOF or a GOF mutation and for a set of background pairs. CNL–CNL: DEG and suppressor have both a copy number loss; CNL–CNG: DEG and suppressor have a copy number loss and gain, respectively. ***P < 0.0005 (Fisher's exact test). **(B)** (left) Fraction of dispensable essential genes with no deleterious mutation across haploid *S. cerevisiae* strains, with a deleterious mutation in at least one of the strains but not co-occurring with deleterious mutations in any of its bypass suppressor genes, and with at least one strain in which it has a deleterious mutation co-occurring with a deleterious mutation in one of its known bypass suppressor genes. (right) Number of dispensable essential genes with a deleterious mutation in any of the haploid *S. cerevisiae* strains co-occurring with a deleterious variant in at least one of its known bypass suppressor genes using the bypass suppression network (pink) and a set of 1,000 randomized networks. In both analyses, only bypass suppression gene pairs with LOF suppressor mutations are considered.

phenotypes were tested (Delneri et al, 2008; Ohnuki & Ohya, 2018). In strains in which a copy of a DEG was lost, we evaluated if the corresponding suppressor gene had a simultaneous copy number change. Interestingly, bypass suppression gene pairs with LOF and GOF suppressor mutations showed different preferences for co-occurring copy number changes, in agreement with their LOF or GOF phenotype. Bypass suppression gene pairs that involved a LOF suppressor mutation were enriched for co-loss of both dispensable essential and suppressor genes (*P* < 0.0005, Fisher's exact tests, Figs 4A and S4A). In contrast, cases with GOF suppressor mutations were enriched for events in which CNL of the DEG was accompanied by a copy number gain of the suppressor gene (*P* < 0.005, Fisher's exact tests, Figs 4A and S4A). Thus, when the DEG has a CNL in a natural strain, the functional effect of the bypass suppressor mutation (GOF or LOF) identifies the most likely copy number change of the suppressor gene in that same strain. Next, we asked whether deleterious coding mutations in DEGs and in identified bypass suppressor genes co-occurred in *S. cerevisiae* isolates. We only considered haploid strains so the deleterious effects of mutations would not be masked by other alleles. When considering only bypass suppression gene pairs in which the suppressor carried a LOF mutation, we found 18 cases in which both the DEG and the suppressor gene carried deleterious mutations in at least one of the haploid strains, significantly more than in randomized gene pairs (*P* < 0.05, randomization test, Fig 4B). As expected, we did not observe a similar enrichment in diploid strains (*P* > 0.05, randomization test, Fig S4B) or for gene pairs involving GOF suppressor mutations (*P* > 0.05, randomization test, Fig S4C). Thus, the bypass suppression network mapped in a laboratory environment reflects evolutionary outcomes in natural *S. cerevisiae* strains.

## Co-occurrence of viability changes and fixed bypass suppressor mutations

We have shown that genes that are dispensable essential in *S. cerevisiae* are often non-essential in other species (Fig 2B). Differences in the genetic background in those species may be responsible for these changes in essentiality. Here, we hypothesized that the genetic changes that bypass the essentiality of a gene in *S. cerevisiae* should be reflected in the genome of species in which

the gene is also dispensable (i.e., non-essential or absent). To test this, we evaluated whether changes in essentiality for DEGs in a given target species co-occurred with bypass suppressor mutations that were fixed in the target genome. Briefly, we considered as equivalent bypass mutations those that could reduce or increase the gene activity in the target species, for LOF and GOF suppressors, respectively (see the Materials and Methods section). Given that genome-scale essentiality data are scarce, we focused our analysis on *S. pombe*, for which high-quality essentiality data are available for most genes (Harris et al, 2022).

We found that 67% (18/27) of the *S. cerevisiae* DEGs that are non-essential in *S. pombe* co-occurred with bypass suppressor mutations in that species, whereas this happened for only 26% (12/47) of the DEGs that were essential in *S. pombe* (*P* < 0.005, Fisher's exact test, Fig 5A and B). A similar trend (48%) was observed for *S. cerevisiae* DEGs that were absent (i.e., without an ortholog) in *S. pombe*, although this difference was not significant compared with the set of essential orthologs (*P* > 0.05, Fisher's exact test, Fig 5B). To increase the statistical power of our analyses, we combined the non-essential and absent genes in *S. pombe* into a single set and observed a clear difference with the essential orthologs (2.3-fold enrichment, *P* < 0.005, Fisher's exact test).

We controlled for potential biases to ensure the robustness of our observation (Fig 5B). We evaluated the effect of interaction degree by generating 1,000 randomized bypass suppression networks while respecting the original topology (Fig S5A) and by considering only DEGs with a single known bypass suppressor (Fig S5B). In addition, we removed bypass suppression interactions from the literature which may have been identified because of phylogenetic properties (Fig S5C), functionally related bypass suppression pairs which may be prone to present similar evolutionary patterns (Fig S5D), and every node in the network to discard dependence on a single gene (Fig S5E). We only considered suppressors with 1:1 orthologs or absent in *S. pombe* to account for the potential expression divergence of duplicated genes (Fig S5F) and calculated the genes with large expression changes between both species to identify gene activity changes (Fig S5G). Also, we applied three alternative orthology mappings (Fig S5H) and used essentiality annotations and orthology mappings from *C. albicans* (Fig S5I). In all these analyses, DEGs without orthologs or with non-essential orthologs more often co-

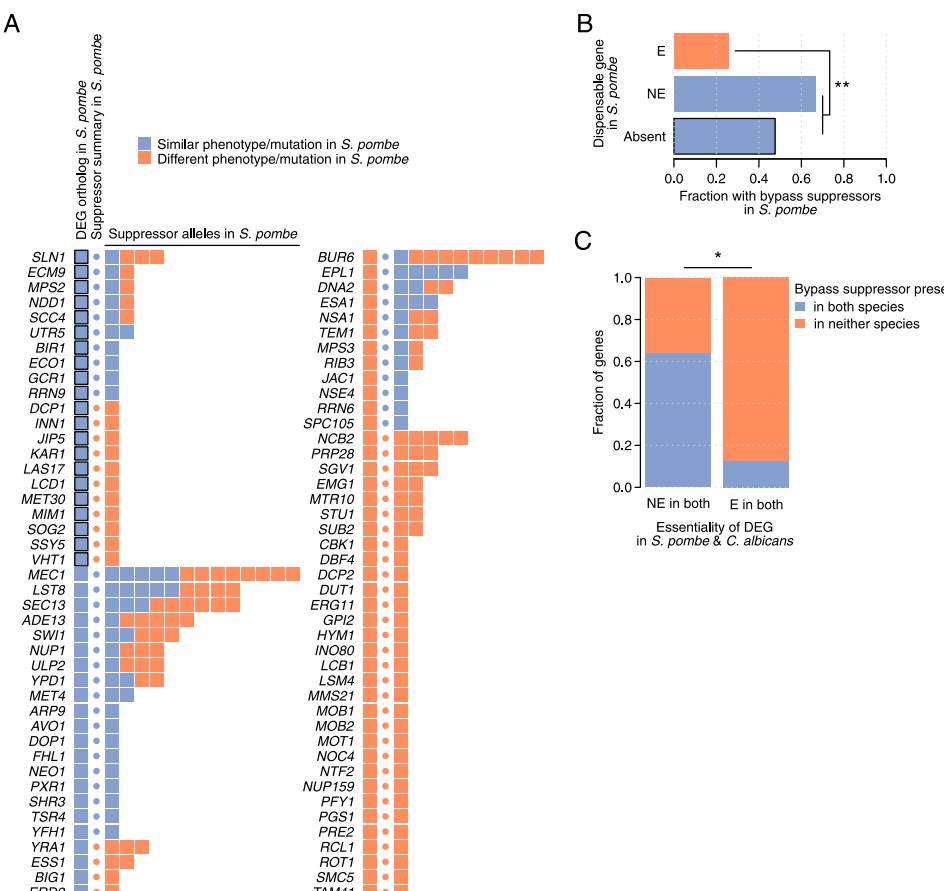

**Figure 5. Changes in essentiality co-occur with bypass suppressor mutations.**
**(A)** Dispensable essential *S. cerevisiae* genes without an ortholog or with a 1:1 ortholog in *S. pombe*, and their bypass suppressors. Color code reflects whether dispensable essential and bypass suppressor genes have similar phenotypes (i.e., absent or non-essential) and mutations, respectively, in *S. pombe* compared with the bypass suppression interactions identified in *S. cerevisiae*. Blue squares with a black border identify dispensable essential genes without an ortholog in *S. pombe*. The circle indicates, for each dispensable essential gene, whether any of the bypass suppressor mutations is present in *S. pombe*. **(B)** Fraction of dispensable essential genes with at least one bypass suppressor mutation in the *S. pombe* genome. Dispensable essential genes are grouped by the phenotype of their 1:1 ortholog in *S. pombe* (E, essential; NE, non-essential; absent: without an ortholog). **(C)** Fraction of dispensable essential genes with bypass suppressor mutations in both *S. pombe* and *C. albicans* or in neither of those species. Dispensable essential genes are grouped by the essentiality of their 1:1 orthologs in those species. **(B, C)** Statistical significance was calculated using Fisher's exact tests. *$P < 0.05$; **$P < 0.005$.

occurred with bypass suppressor mutations than DEGs with essential orthologs ($P < 0.05$, Fisher's exact tests). Conversely, switching LOF and GOF annotations resulted in a non-significant difference, as expected ($P > 0.05$, Fisher's exact test, Fig S5J).

Finally, we selected DEGs with 1:1 orthologs in both *S. pombe* and *C. albicans* and found that DEGs with non-essential orthologs in both species were more likely to have bypass suppressor mutations in those species than DEGs with essential orthologs ($P < 0.05$, Fisher's exact test, Fig 5C). In all, these analyses reveal that the relationship between DEGs and their bypass suppressor genes identified in *S. cerevisiae* is reflected in the gene essentiality and mutational space of other species.

# Discussion

Differences between essential and non-essential genes have been widely characterized (Figs 1C and S2G and I) and a myriad of machine learning algorithms have exploited this information for the successful prediction of gene essentiality (Hwang et al, 2009; Lloyd et al, 2015; Zhang et al, 2016). Recently, we and others have identified a subset of *S. cerevisiae* essential genes that become dispensable

in the presence of specific genetic variants (Liu et al, 2015; van Leeuwen et al, 2020). Here, we have combined these datasets of DEGs, after showing they exhibit similar properties (Fig 1), for the comprehensive characterization of these genes. While recapitulating previously reported features in smaller datasets, we have also revealed new properties of DEGs (Figs 1C and 2). These features can be incorporated in existing methods for the prediction of essential gene dispensability (van Leeuwen et al, 2020). Because properties of DEGs are highly conserved (van Leeuwen et al, 2020), predictions could potentially target other species. Although the differences between dispensable essential and core essential genes resemble the differences between essential and non-essential genes (Figs 1C, 2B, and S2I), dispensable essential and non-essential genes also make up two clearly distinct groups (Figs 1C and S2H). Thus, in contrast to the classical binary classification of genes based on their essentiality, three different sets of genes exist with specific properties that distinguish them from each other: non-essential, dispensable essential, and core essential genes, as was also previously suggested (Liu et al, 2015).

Importantly, we presented extensive evidence of the distinct evolutionary pressure exerted on these gene sets by performing phylogenetic analyses spanning very different evolutionary timescales (Figs 2 and S2), further expanding previous observations (Liu et al, 2015;

van Leeuwen et al, 2020). The observed differences in conservation of dispensable essential compared with core essential *S. cerevisiae* genes in *S. uvarum*, *C. albicans*, *S. pombe*, and even human, which diverged from *S. cerevisiae* ~1 billion years ago, reflect the substantial evolutionary footprint of essential gene dispensability.

For a better characterization of the mechanisms associated with the tolerance of highly deleterious mutations, we integrated data from multiple studies to build a bypass suppression interaction network between DEGs and their suppressors. Several properties emerged demonstrating the modularity and structure of the bypass suppression network. Complexes tended to be either composed of only dispensable essential subunits or of only core essential subunits (Fig S1B), mirroring the essentiality composition bias previously described (Hart et al, 2007) and the functional modularity that complexes encapsulate. Dispensable essentiality, thus, would be a modular feature of protein complexes (Li et al, 2019b), as is essentiality. Also, protein complexes exhibited monochromaticity of suppressor type (Fig 3C) with members of the same complex being all suppressed by either LOF or GOF mutations. Last, members of the same complex exhibited interaction coherence, with cocomplexed DEGs sharing suppressors and cocomplexed suppressor genes interacting with the same DEGs (Fig 3D), as illustrated in Figs 3E and S3G. All these observations expose the inherent modularity of the bypass suppression network and suggest that similar suppression mechanisms apply for functionally related genes, which can lead to the identification of new dispensable essential and suppressor genes. Certainly, network modularity is not restricted to the bypass suppression network, and it is in fact a hallmark of a global genetic interaction network (Costanzo et al, 2016), but it is particularly relevant here, given its directionality, small size, and low interaction density, reflecting the strong functional relationships bypass suppression interactions encapsulate.

The potential role of genetic suppression in explaining the existence of deleterious variants among natural populations (Chen et al, 2016) is still not fully understood. To address this knowledge gap, we evaluated how bypass suppression gene pairs reflected simultaneous genomic changes across evolution. Remarkably, we found co-occurrence of copy number changes and deleterious mutations in both the dispensable essential and the suppressor genes across *S. cerevisiae* strains (Fig 4). Furthermore, *S. cerevisiae* DEGs that were absent or non-essential in *S. pombe* were more likely to co-occur with a bypass suppressor mutation in the *S. pombe* genome than DEGs that were essential in *S. pombe* (Fig 5). These results suggest that within- and across species genetic variability can follow the same evolutionary paths as spontaneous mutations in a laboratory environment, illustrating the constraints genetic networks may impose on evolutionary trajectories.

# Materials and Methods

## Dispensable essential gene analyses

### Dispensable essential gene datasets
We retrieved DEGs in *S. cerevisiae* from two systematic experimental datasets (Liu et al, 2015; van Leeuwen et al, 2020) and from a study

that compiled data from the literature (van Leeuwen et al, 2020). Cases of bypass suppression were identified by looking for rare survivors in populations of 100–150 million cells deleted for an essential gene (van Leeuwen et al, 2020; experimental dataset), by following germination of single-deletion mutant spores (Liu et al, 2015), or by a mixture of methods, including directly testing the effect of a mutation on strain viability (van Leeuwen et al, 2020; literature dataset). Because mutation rates are generally low and specific point mutations that can bypass an essential gene are thus unlikely to arise within a single spore, the Liu et al study mainly identified cases of suppression that involved changes in chromosome number, which occur more frequently than point mutations. In contrast, the van Leeuwen et al (2020) study identified essential genes that were bypassed by ploidy changes and by single nucleotide changes in the genome, and as a result identified a higher number of DEGs. The set of tested genes are explicitly mentioned in the systematic studies, whereas for the literature set they are unknown and, therefore, we used all essential genes in *S. cerevisiae*. The combined dataset contained the DEGs identified in any of the three individual datasets. As tested genes, we considered all tested genes in the systematic studies and the dispensable genes identified in the literature set. We randomly generated 1,000 sets of genes of the same sizes as the individual datasets, sampling from the corresponding set of tested genes.

We calculated the overlap between the different datasets by counting the number of dispensable genes found across two and three datasets (Fig 1A). We repeated the same process in the randomly generated datasets to derive empirical *P*-values.

### Essentiality data
In our analyses, we used essentiality data from *S. cerevisiae* (van Leeuwen et al, 2020), *S. uvarum* (Sanchez et al, 2019), *C. albicans* (Segal et al, 2018), *S. pombe* (downloaded in November 2021 from PomBase [Harris et al, 2022]), and human cell lines (Hart et al, 2015). We considered human essential genes those that were required for viability in at least three of the five cell lines tested. In *C. albicans*, genes with essentiality confidence scores above 0.5 were classified as essential and the remaining genes as non-essential.

### Orthology mappings
We used PantherDB 16.1 (Mi et al, 2021) to identify orthology relationships (Figs 2, 5, S2, and S5). When indicated, we also used OrthoMCL (Li et al, 2003), SonicParanoid (Cosentino & Iwasaki 2019), based on the popular InParanoid (Sonnhammer & Östlund, 2015), and PomBase (Wood et al, 2012) orthology mappings.

### Functional enrichment of dispensable essential genes
For each DEG set and each of the 14 broad functional classes previously defined (Costanzo et al, 2016), we calculated the fold enrichment as the fraction of DEGs annotated to that functional class with respect to the corresponding fraction of core essential genes (Fig S1A). We calculated the statistical significance with two-sided Fisher's exact tests and corrected for the multiple tests across the 14 functional classes using the FDR. We considered the cases with a $P < 0.05$ and FDR < 10% as significant enrichments and depletions.

### Enrichment for non-essential genes in the Sigma1278b strain

For each dispensable gene set, we calculated the fold enrichment as the ratio of DEGs identified as non-essential in the Sigma1278b strain divided by the analogous ratio of core essential genes (Fig S1D). *P*-values were calculated using two-sided Fisher's exact tests.

### Complex dispensability bias

For each DEG set, we counted the number of complexes (Meldal et al, 2021) in which all essential subunits were identified either as dispensable or core essential genes. We repeated the same process using the randomly generated datasets to derive empirical *P*-values (Fig S1B).

### Properties of dispensable essential genes

We queried a panel of 17 numeric and four binary features to characterize the set of DEGs (Fig 1C). Protein-level numeric properties included abundance (Ho et al, 2018), half-life (Belle et al, 2006), degradation rate (Christiano et al, 2014), number of structural domains identified by Pfam 34.0 (Mistry et al, 2021), fraction of structurally disordered residues calculated by VLS2b (Peng et al, 2006) downloaded from d2p2.pro (Oates et al, 2013), sequence length, and number of protein–protein interactions degree (Koch et al, 2012). For members of protein complexes (Meldal et al, 2021), we counted the number of different complexes in which they were found and calculated their cocomplex degree as the number of protein partners present in those complexes. Gene-level numeric properties included transcript count (Lipson et al, 2009), expression variance under different environmental conditions (Gasch et al, 2000), coexpression degree calculated as the number of genes with similar expression profiles (i.e., MEFIT scores > 1) (Huttenhower et al, 2006), multifunctionality calculated as the number of GO SLIM (Ashburner et al, 2000) biological process annotations downloaded from SGD (Cherry et al, 2012), number of paralogs (i.e., copy number) (Koch et al, 2012), ratio of non-synonymous to synonymous substitutions (dN/dS) to quantify sequence evolution (Koch et al, 2012), and number of yeast (i.e., yeast conservation) and distant (i.e., broad conservation) species in which the gene is conserved (Koch et al, 2012). We also defined four binary features to describe if proteins had a structural domain (Mistry et al, 2021), localized to a membrane (Babu et al, 2012), or belonged to a protein complex (Meldal et al, 2021), or if genes had any duplicate according to YeastMine (Cherry et al, 2012).

For each numerical feature, values that corresponded to DEGs were z-score normalized using the median and SD of the values that corresponded to core essential genes. A resulting positive median z-score identifies a feature in which DEGs tend to have higher values than core essential genes. Conversely, a negative median z-score identifies a feature in which DEGs tend to have lower values than core essential genes. Dot size in plots is proportional to the absolute median z-score value and the dot color identifies the set of genes with higher feature values. We calculated the statistical significance by means of Mann–Whitney *U* tests. For each binary feature, we calculated the fold enrichment as the ratio of DEGs with that particular feature divided by the equivalent ratio for core essential genes. Dot size is proportional to the absolute value of the $\log_2$ of the fold enrichment and the dot color identifies the set of genes with a higher ratio for a particular feature. We calculated the

*P*-values with two-sided Fisher's exact tests. We corrected for the multiple tests across the 21 features by calculating the FDR. All shown dots correspond to features with significant differences (*P* < 0.05, Mann–Whitney *U* test or Fisher's exact test, and FDR < 10%) between the gene sets. For visualization purposes, all significant enrichments with median z-scores or fold enrichment values below 0.1 were shown with dots of the same size. We followed the same approach to characterize (i) dispensable essential versus non-essential genes (Fig 1C); (ii) essential versus non-essential genes (Fig 1C); (iii) DEGs with multiple suppressors versus DEGs with a single suppressor (Fig S3F).

### Analyses on S. cerevisiae strains

We downloaded gene presence/absence data for a large panel of *S. cerevisiae* strains (Li et al, 2019a) and defined several core pan-genome gene sets at different stringency levels (see x-axis in Fig S1E). For instance, a threshold of 10 identifies the core pangome composed of all genes, absent only in 10 strains or less. For each DEG dataset and pangenome, we calculated the fraction of DEGs missing from the pangenome and the corresponding fraction for core essential genes, from which we calculated the fold enrichment. We also calculated fold enrichments for core essential genes versus the complete set of DEGs and for the non-essential genes versus essential genes. *P*-values were calculated with two-sided Fisher's exact tests.

We retrieved precomputed LOF data for *S. cerevisiae* strains (Peter et al, 2018) from http://1002genomes.u-strasbg.fr/files/, including frameshift mutations and missense mutations predicted to be deleterious by SIFT (Ng & Henikoff, 2001). We calculated the number of strains in which these mutations affected each gene and aggregated the results per gene set (i.e., dispensable essentials, core essentials, and non-essentials). *P*-values were calculated using two-sided Fisher's exact tests (Fig S2A).

For each strain, we counted the genes affected by CNL events in a panel of *S. cerevisiae* strains (Peter et al, 2018) and aggregated the result per gene set (Fig S2C). *P*-values were calculated using two-sided Fisher's exact tests. Finally, we retrieved dN/dS data for the same panel of *S. cerevisiae* strains and grouped them by gene set (Fig S2B). *P*-values were calculated using Mann–Whitney *U* tests.

### Orthology relationships of dispensable essential genes

For each gene, we calculated its orthology relationships in *C. albicans*, *S. pombe*, and human (Figs 2A and B and S2D and E). Specifically, we considered gene absence, gene duplication (including 1:N and N:M orthology relationship), N:1 relationships, and 1:1 orthologs. For 1:1 orthologs, we evaluated the essentiality in the target species. For each species and property, the fold enrichment was calculated as the fraction of DEGs with respect to the fraction of core essential genes with that property. *P*-values were calculated by two-sided Fisher's exact tests. We used the same approach to compare dispensable essential to non-essential genes (Fig S2H) and non-essential to essential genes (Fig S2G and I).

### Gene age

For each gene, we calculated its age by identifying the farthest species from *S. cerevisiae* with a present ortholog. We used orthology relationships for 98 species from PantherDB (Mi et al,

2021). The phylogenetic tree to calculate species relationships was downloaded from UniProt (UniProt Consortium, 2021), and for each species, we calculated the distance to *S. cerevisiae* as the number of main branches separating them. Thus, genes with age 0 are specific to *S. cerevisiae* and not present in any other of the 98 species, whereas age five corresponds to genes present in the most distantly related species. We grouped gene ages for each gene set (core, dispensable, and non-essentials) and calculated *P*-values with Mann–Whitney *U* tests (Fig 2C).

### Gene loss

For each gene of age X, we calculated the fraction of species closer to *S. cerevisiae* (distance < X) in the phylogenetic tree with that gene absent from their genome. For instance, for a given gene of age 3, we calculated the fraction of species at distance 1 or 2 to *S. cerevisiae* with the gene of interest absent. We aggregated data for each gene set (core essentials, dispensable essentials, and non-essentials) and calculated *P*-values by means of Mann–Whitney *U* tests (Fig 2E).

### Cancer cell lines

We used fitness data from genome-scale CRISPR–Cas9 knockout screens in 1,070 cancer cell lines from DepMap (Meyers et al, 2017). For each gene, we calculated the median effect of gene knockout on cell proliferation and the associated SD across all cell lines. *P*-values were calculated using Mann–Whitney *U* tests (Fig 2F).

### Sequence analysis

For all 1:1 ortholog pairs between *S. cerevisiae* and *S. pombe*, we calculated their protein sequence identity (Fig 2G). Sequence length similarity was calculated as the length ratio between the shortest of the sequences with respect to the longest (Fig S2J). Thus, values closer to one describe sequence pairs of similar length, whereas values closer to 0 correspond to sequences of very different lengths. *P*-values were calculated using Mann–Whitney *U* tests. We followed the same approach to compare *S. cerevisiae* and *C. albicans* sequences (Fig S2K and L).

## Suppression network analyses

### Interaction data

We combined suppression interactions from our recent study (van Leeuwen et al, 2020) with interactions found in the literature (van Leeuwen et al, 2020) including only deletions of essential genes suppressed under standard conditions. We generated 1,000 randomized networks respecting the topology (i.e., maintaining the total number of connections of each gene) using the BiRewire R package (Iorio et al, 2016). We calculated the number of bypass suppression pairs present in both datasets and compared that value to the number of overlapping pairs in randomized networks to derive an empirical *P*-value (Fig 3A).

### Functional overlaps

We calculated the fraction of bypass suppression gene pairs that coded for proteins localized to the same subcellular compartment (Huh et al, 2003), had MEFIT (Huttenhower et al, 2006) coexpression scores above 1.0, were annotated to the same biological process GO term (Myers et al, 2006; Costanzo et al, 2016), and coded for

members of the same complex (Meldal et al, 2021) and molecular pathway (Kanehisa et al, 2016). We repeated this calculation with the non-interacting gene pairs in the bypass suppression network and derived fold enrichments and *P*-values using two-sided Fisher's exact tests. We applied this approach to the individual and the combined datasets (Fig S3A).

### Complex monochromaticity by suppression mode

We selected a non-redundant set of 17 protein complexes with at least two dispensable essential subunits in the bypass suppression network. We only kept one representative complex when several complexes had the same set of DEGs. For each complex, we calculated if all dispensable essential subunits could be suppressed by the same suppressor mode (LOF or GOF). Note that in one complex, all subunits could not only be suppressed by LOF suppressors but also by GOF suppressors (indicated by "LOF & GOF" in the panel). We counted all complexes with this monochromaticity in suppression mode and compared that value to the number of monochromatic complexes in a set of 1,000 randomized bypass suppression networks to derive an empirical *P*-value (Fig 3C). We applied the same approach to the two individual suppression networks to discard a bias in the literature dataset (Fig S3D).

### Network modularity based on cocomplex relationships

We counted the number of DEGs within the same protein complex (Meldal et al, 2021) that shared at least one suppressor. We repeated the same calculation using pairs of DEGs belonging to different complexes to derive fold enrichment and a *P*-value calculated with a two-sided Fisher's exact test. We followed the same approach querying for interactors of bypass suppressors instead of the interactors of DEGs (Fig 3D).

### Functional preferences

We annotated the dispensable essential and suppressor genes in the network using 14 broad functional classes previously defined (Costanzo et al, 2016). We then calculated the number of bypass suppression gene pairs within each pair of classes and repeated the process in 1,000 randomized bypass suppression networks to derive empirical *P*-values. We used the median frequencies in the randomized set to calculate the fold enrichments (Fig S3B, top). Only fold enrichments of significant associations are shown (*P* < 0.05 and FDR < 10%).

We calculated fold enrichments for suppressors as the fraction of those genes in each class with respect to the corresponding fraction of background genes (Fig S3B, bottom). We calculated *P*-values using two-sided Fisher's exact tests and corrected for multiple testing using the FDR.

### Overlap with genetic interactions

To evaluate the overlap between bypass suppression interactions and genetic interactions, we used a global genetic interaction map (Costanzo et al, 2016), which includes data for most gene pairs in *S. cerevisiae*. LOF alleles of the gene pairs were screened using hypomorphic and deletion alleles to query essential and non-essential genes, respectively. We used the standard cutoffs to identify negative and positive genetic interactions. For gene

pairs screened more than once (for instance, using different hypomorphic alleles of the same gene), we implemented a consensus approach in which we considered the gene pair to genetically interact if 50% or more of the corresponding allele pairs showed an interaction. We split the bypass suppression interaction pairs by their suppression mode and counted for each set the number of overlapping pairs with negative and positive genetic interactions. Next, to calculate the expected overlap by chance with genetic interactions, we generated a background set including all possible DEG and suppressor gene pairs, and removed the pairs present in the bypass suppression interaction network. Then, we calculated the overlap of this background gene pairs with negative and positive interactions. We compared the corresponding ratio to the overlaps obtained using the bypass suppression interaction network, and calculated the statistical significance by two-sided Fisher's exact tests (Fig 3B).

### Clustering of genetic interaction profiles

We downloaded the genetic interaction profile similarities computed using the complete genetic interaction profiles from https://thecellmap.org/ (Usaj et al, 2017). For genes with multiple alleles, we averaged the similarity values across alleles. We performed hierarchical clustering of the genetic interaction profiles using the R function *hclust*, and used a cutoff of 0.2 to define a network of genes with high-profile similarities (Fig 3F).

### Agreement in copy number changes and suppression mode across *S. cerevisiae* strains

We defined CNL and copy number gain (CNG) events as having a copy number below 1 or above 1, respectively, per haploid genome as defined by the 1,011 genomes project (Peter et al, 2018). The study used a combination of flow cytometry, sequencing coverage analysis across 1-kb windows, and allele frequency distributions to determine copy number changes, including both aneuploidies and segmental duplications (Peter et al, 2018). For each bypass suppression gene pair, we calculated the number of strains in which both genes had a CNL (i.e., co-loss events, CNL–CNL) and a CNL event for the DEG and CNG for the suppressor gene (i.e., loss-gain events, CNL–CNG). We disregarded 18 hypermutated strains with copy number changes in >33% of the genes in the bypass suppression network and aggregated co-loss and loss-gain events for all bypass suppression gene pairs after splitting pairs by their suppression mode (LOF or GOF). We repeated the same calculation with a background set of gene pairs composed of all possible DEG-suppressor pairs, after removing the gene pairs in the bypass suppression network. We compared the proportion of co-loss versus loss-gain events for the LOF and GOF bypass suppression pairs, for LOF bypass suppression pairs and background pairs, and for GOF bypass suppression pairs and background pairs (Fig 4A). We calculated the statistical significance by two-sided Fisher's exact tests. Finally, we followed the same approach to count the number of strains in which each set (LOF and GOF bypass suppression and background pairs) overlapped more often with co-loss than loss-gain events (and vice versa). We compared the resulting proportions as explained above and calculated the statistical significance with two-sided Fisher's exact tests (Fig S4A).

### Co-mutation in *S. cerevisaie* strains

We defined as deleterious mutations the missense mutations predicted as damaging by SIFT, indel mutations, and frameshift mutations (see the section "Analyses on *S. cerevisiae* strains" above). For each DEG, we retrieved the strains in which it had a deleterious mutation and checked if any of its bypass suppressor genes was also mutated in any of those strains. We counted the number of DEGs co-mutated in any strains with any of their suppressor genes, the number of DEGs mutated alone, and the number of DEGs not mutated in any strain. We repeated the same process using 1,000 randomized bypass suppression networks. We performed this calculation using (1) LOF bypass suppression pairs and haploid strains (Fig 4B); (2) LOF bypass suppression pairs and diploid strains (Fig S4B); and (3) GOF bypass suppression pairs and haploid strains (Fig S4C).

### Phenotypic changes across species and presence of bypass suppressor mutations

We hypothesized that the relationships between DEGs and bypass suppressor mutations identified in *S. cerevisiae* should be reflected in the evolutionary landscape of other species. To test this hypothesis, we identified DEGs that were non-essential or absent in a given target species and evaluated if the bypass suppressor mutations were fixed in the given target genome. To determine if a bypass suppressor mutation was fixed in another species, we took into account the effect of the suppressor mutation on gene function. Briefly, for LOF suppressors, we evaluated if mutations in the target species would reduce the gene activity with respect to *S. cerevisiae*. Conversely, for GOF suppressors, we evaluated if the mutations would increase the gene activity.

First, we annotated the orthology relationship of each DEG in *S. pombe*. We only considered DEGs absent in *S. pombe* or with a 1:1 ortholog. For genes with 1:1 orthologs, we annotated the essentiality of the ortholog in that species. We also annotated the orthology relationships of bypass suppressor genes in *S. pombe*. For suppressors with 1:1 orthologs, we performed a sequence alignment between the protein sequences of both species.

We next describe the set of rules that we evaluated to identify cases with equivalent bypass mutations in *S. pombe*. Briefly, in LOF suppressors, we looked for orthologs with decreased activity with respect to the *S. cerevisiae* gene, whereas in GOF suppressors, for orthologs with increased activity. The first set of rules was based on orthology relationships. We considered *S. pombe* to have a LOF bypass mutation if the suppressor gene was absent. Also, if it had an N:1 ortholog, which could be similar to a copy number decrease and, thus, a decrease in activity. Suppressors with more than one ortholog in *S. pombe* or with a 1:1 ortholog were considered non-equivalent LOF bypass mutations because their copy number did not decrease. Conversely, we considered as GOF bypass mutations cases in which the suppressor gene had more than one ortholog, similar to increasing their copy number and their activity, and non-equivalent GOF bypass mutations cases in which there was a N:1, 1:1, or absent ortholog in *S. pombe*.

The second set of rules we used to evaluate equivalent mutations was based on protein sequences. We only considered frameshift, nonsense, and missense mutations of suppressor genes with 1:1 orthologs. For the rest of cases, only the orthology rules (see

above) were applied. The position of the nonsense and frameshift suppressor mutations identifies the part of the protein that should remain functional. Functionality encoded beyond that residue is compromised. Thus, we considered 1:1 orthologs in *S. pombe* with a shorter sequence than the position of the nonsense or frameshift suppressor mutation as LOF bypass mutations. Conversely, we considered cases in which the ortholog sequence was equal or longer than the position of the nonsense or frameshift suppressor mutation as non-equivalent LOF bypass mutations. In cases with missense mutations, we performed a sequence alignment between the *S. cerevisiae* suppressor gene and its 1:1 ortholog in *S. pombe*. We considered the ortholog to have an equivalent LOF bypass mutation if the same mutated residue or a gap was found in the aligned mutated position of the ortholog sequence. If the aligned mutated residue was the same as in the WT *S. cerevisiae* sequence (i.e., unmutated), we considered the ortholog to have a non-equivalent LOF bypass mutation. Cases in which the aligned position had different residues in *S. pombe* (not the WT and not the suppressor mutation) could not be classified as either equivalent or non-equivalent LOF bypass mutations. For GOF suppressors with a missense mutation and a 1:1 ortholog in *S. pombe*, we also performed a sequence alignment between the suppressor gene and its 1:1 ortholog. We considered the ortholog to have an equivalent GOF bypass mutation if the same mutated residue was found in the aligned mutated position of the ortholog sequence. If the aligned mutated residue was the same as in the WT *S. cerevisiae* sequence (i.e., unmutated), we considered the ortholog to have a non-equivalent GOF bypass mutation. The rest of cases could not be classified as either equivalent or non-equivalent GOF bypass mutations. We also evaluated missense mutations of suppressors with unknown suppression mode that had 1:1 orthologs in *S. pombe*. Cases with the exact same mutation in the ortholog were classified as equivalent bypass mutations, whereas cases in which the residue did not change in the 1:1 ortholog were classified as non-equivalent bypass mutations. The remaining suppressors with unknown suppression mode were not evaluated. Importantly, in suppressor genes with a frameshift, nonsense, or missense mutation, and with a 1:1 ortholog in *S. pombe*, the sequence-based assessment took precedence over the orthology based evaluation.

Finally, we considered a DEG to have an equivalent bypass suppressor in *S. pombe* if any of its suppressors satisfied that criteria. We grouped DEGs by their essentiality in *S. pombe*, expecting DEGs with equivalent phenotypes in *S. pombe* (i.e., absent or 1:1 non-essential orthologs) to have equivalent bypass suppressors more often than DEGs with a 1:1 essential ortholog. We calculated the fraction of genes with equivalent bypass suppressors for both gene sets to derive fold enrichment and the *P*-value with a one-sided Fisher's exact test. We compared the fold enrichment of the bypass suppression network to a set of randomized bypass suppression networks, which we used to derive an empirical *P*-value.

We repeated the exact same process (i) using *C. albicans* sequences, orthology relationships, and essentiality annotations; (ii) using orthoMCL (Li et al, 2003), SonicParanoid (Cosentino & Iwasaki, 2019), and PomBase (Wood et al, 2012) as alternative orthology mappings; (iii) considering only DEGs with a single bypass suppressor to control for the bias introduced by gene degree; (iv)

removing bypass suppression pairs from the literature which may have been potentially identified by phylogenetic approaches; (v) removing cocomplex and copathway bypass suppression pairs which may be more prone to present similar phylogenetic patterns; (vi) switching LOF and GOF annotations to demonstrate the specificity of our sets of rules; (vii) removing every node in the network to discard dependence on a single gene.

### On the use of orthology relationships to identify gene activity changes between species

Genes duplicated in the *S. pombe* lineage but not in *S. cerevisiae* should result in 1:N orthology relationships (one gene in *S. cerevisiae*, N genes in *S. pombe*). At the moment of the duplication event, there is usually an increase in transcript and protein levels (Pavelka et al, 2010) causing an increase in gene activity. To evaluate if the initial increase in expression of duplicated genes is maintained in *S. pombe*, we quantified expression changes between *S. cerevisiae* (Lipson et al, 2009) and *S. pombe* (Grabherr et al, 2011; Koch et al, 2012), which highly correlate for 1:1 orthologs (r = 0.62, *P* < 0.0005, Fig S5K), by aggregating expression levels of orthology groups. For instance, in 1:N orthology relationships, we compared the expression level of one *S. cerevisiae* gene to the result of adding the expression levels of the N genes in *S. pombe*. In N:1 orthology relationships, we compared the aggregated expression levels of the N *S. cerevisiae* genes to the expression level of a single gene in *S. pombe*. In N:M orthology relationships, we compared the combined expression levels of N genes in *S. cerevisiae* to the aggregated expression level of M genes in *S. pombe*. In 1:1 orthologs we compared the expression levels of the ortholog in each species. For each orthology relationship, we calculated the expression ratio by dividing the expression level of the gene/s in *S. pombe* by the expression level of the gene/s in *S. cerevisiae*. In agreement with the initial copy number event, expression ratios were higher for 1:N orthology relationships than for 1:1 orthologs (median values 1.345 and 0.791, respectively, *P* < 0.0005, Mann–Whitney *U* test, Fig S5L). Conversely, expression ratios were lower for N:1 than for 1:1 orthology relationships (median = 0.568, *P* < 0.0005, Mann–Whitney *U* test). That is, for each gene transcript in *S. cerevisiae*, there were more transcripts in *S. pombe* for the aggregated 1:N orthologs than for 1:1 orthologs, and fewer transcripts for N:1 orthologs than for 1:1 orthologs. Therefore, in general, the expression level change resulting from a copy number gain or loss is kept in *S. pombe*.

To account for cases in which the initial copy number changes and the expression changes in *S. pombe* may not agree, we implemented an alternative approach to identify genes with increased or decreased activity between *S. cerevisiae* and *S. pombe*, by replacing the use of orthology relationships (see the "Phenotypic changes across species and presence of bypass suppressor mutations" section) with the quantification of expression changes between both species. For all genes with identified orthologs, we calculated the *S. pombe* to *S. cerevisiae* expression ratio between both species as explained above. Genes with an extreme decrease or increase in expression levels between orthologs (genes within the bottom 5% and top 5% expression ratios, respectively) were labeled as genes with decreased and increased activity, respectively. We considered a LOF bypass mutation equivalent in *S. pombe* if the suppressor gene was within the bottom 5% expression ratios

and non-equivalent otherwise. Conversely, we considered a GOF bypass mutation equivalent in *S. pombe* if the suppressor gene was within the top 5% expression ratios and non-equivalent otherwise. By combining expression ratios with the rules based on protein sequences and gene absence (see the "Phenotypic changes across species and presence of bypass suppressor mutations" section), we calculated if DEGs absent or non-essential in *S. pombe* were more likely to have the corresponding bypass suppressors in that species, as explained in the previous section.

However, functional divergence may take place after the duplication event and the initial increase in gene activity may be modulated, which is difficult to evaluate without gene-specific functional assays. To control for the impact to our results of the potential functional divergence after copy number changes, we repeated the same analysis shown in Fig 5B but considering only suppressors with a 1:1 ortholog in *S. pombe* or absent in that species. Thus, we did not consider orthology relationships (i.e., duplicated suppressors or with N:1 orthologs) or transcript changes to identify equivalent GOF or LOF bypass suppressors in *S. pombe*.

# Supplementary Information

# Acknowledgements

This work was supported by an Eccellenza grant from the Swiss National Science Foundation (PCEGP3_181242) (J van Leeuwen) and a Ramon y Cajal fellowship (RYC-2017–22959) (C Pons).

## Author Contributions

C Pons: conceptualization, data curation, formal analysis, supervision, visualization, methodology, and writing—original draft, review, and editing.
J van Leeuwen: data curation, visualization, and writing—review and editing.

## Conflict of Interest Statement

The authors declare that they have no conflict of interest.

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
