## [Reviewer comments · Life Science Alliance]

Life Science Alliance

Meta-analysis of dispensable essential genes and their interactions with bypass suppressors

Carles Pons and Jolanda van Leeuwen

DOI: <https://doi.org/10.26508/lsa.202302192>

Corresponding author(s): Carles Pons, Institute for Research in Biomedicine

Review Timeline:

Submission Date:	2023-05-31
Editorial Decision:	2023-06-30
Revision Received:	2023-09-29
Editorial Decision:	2023-10-19
Revision Received:	2023-10-24
Accepted:	2023-10-25

Transaction Report:

June 30, 2023

Re: Life Science Alliance manuscript #LSA-2023-02192-T

Carles Pons
The Barcelona Institute for Science and Technology
Spain

Dear Dr. Pons,

Thank you for submitting your manuscript entitled "Meta-analysis of dispensable essential genes: phylogenetic properties and co-occurrence with bypass suppressor mutations across species" to Life Science Alliance. The manuscript was assessed by an expert reviewer, whose comments are appended to this letter. We invite you to submit a revised manuscript addressing the Reviewer comments.

When submitting the revision, please include a letter addressing the reviewer comments point by point.

Thank you for this interesting contribution to Life Science Alliance. We are looking forward to receiving your revised manuscript.

Sincerely,

B. MANUSCRIPT ORGANIZATION AND FORMATTING:

Reviewer #1 (Comments to the Authors (Required)):

In "Meta-analysis of dispensable essential genes: phylogenetic properties and co-occurrence with bypass suppressor mutations across species", Pons and van Leeuwen integrate multiple functional and genomic datasets to explore the properties of previously identified dispensable essential genes. While I have comments on the analysis work that was performed, one of my main concerns with the manuscript has more to do with how the results and methods are presented. As it is currently written, the paper cannot be understood without being intimately familiar with at least 5-6 other manuscripts. In that regard:

1. Please take the time to properly define all terms used in the main text, methods and figure legends. For example, what is the difference between expression variance (g) and expression variance (e) in Figure 1? Some of these properties and where they were obtained are defined only in the methods of van Leeuwen et al, 2020a and b, while for others there is simply no information available.

2. The same is true of genetic interaction screening terminology. Please take the time to explain what you mean by positive or negative genetic interactions and the differences in methodology between the different experiments and datasets reported in the manuscript. While the authors show that the different datasets include genes with similar properties, experimental considerations might also play a role in determining whether the different approaches are expected to yield similar results.

3. Finally, please improve the methods to make them self-supporting. Not every analysis step from previous manuscripts from which datasets originate needs to be re-written, but the rationale and thresholds used should at least be presented so that reading 5-6 different papers is not required to understand what was done here.

Regarding the analysis and conclusions presented:

4. The paper examines many properties associated with dispensable genes, which appear to add up to many potential associations being tested. Yet, there is no mention of strategies to control for multiple testing in the methods or the main text. This is further obscured by the fact that the number of associations tested is often not explicit. For example, the authors used the same panel of gene features as in van Leeuwen 2020, but failed to redescribe it in detail here so the total number of properties tested here is unclear. Furthermore, in many cases, associations are only categorized as significant or not, without any test statistics or p-values provided. This is in my opinion one of the most significant weaknesses of the manuscript, as the associations of the study fully depend on the robustness of the associations uncovered. Please provide a complete list of associations tested and the strategies used to mitigate multiple hypothesis testing (e.g.: FDR, Bonferonni or other).

5. The level of evidence used to determine if a mutation results in a LOF or GOF in the case of a missense mutation appears very low. Yet, mutations that were classified using SIFT are presented in the text as strictly deleterious or gain of function, without highlighting the fact this is a prediction and providing uncertainty estimates. The accuracy of SIFT can also vary wildly between different proteins, depending on the number of sequences in the input alignment and the protein itself. The authors should provide more information on the confidence associated with their predictions.

6. Furthermore, algorithms based on sequence conservation are much better at classifying LOF mutations compared to GOF. There is also nothing preventing a GOF mutation from being non-conservative and thus classified as a LOF by SIFT. The authors fail to mention these limitations, which have significant implications for several downstream analyses. Factoring in the high potential for false positives in this analysis might for example change the conclusion of the analysis shown in figure 4 and figure 5.

7. When discussing CNL and CNG events, the author should explain a bit better how these events were detected in the Peters 2018 dataset. Are only segmental duplications included, or also CNVs due to aneuploidies?

8. Also regarding CNL and CNG, the authors should explain and support their reasoning as to the effect of duplication on gene activity. Only a small fraction of yeast genes are haploinsufficient, and many of them have relatively flat expression fitness functions (Keren et al 2016, Cell). As the vast majority of natural strains included in Peters et al 2018 are diploids (n=694 diploids out of), shifts to a copy number of 0.5 or 2 might rarely have an effect on fitness, regardless of any potential suppression.

9. I strongly disagree that an N:1 orthology relationship is equivalent to a copy number decrease. This hypothesis disregards any possible functional or expression level divergence that could have occurred. The same is true of the opposite, where authors considered 1:N orthology relationships as GOF. The authors should provide convincing evidence to support this reasoning or remove this aspect of the analysis.

10. The approach used for the identification of suppressor mutations in orthologs completely disregards the potential for cis epistasis that would alter the effect of a potential suppressor mutation. When combined with the probable high false positive rate of the variant effect prediction used by the authors this analysis seems unlikely to be unreliable.

11. General point for figures: please add the number of data points part of each boxplot, and please consider showing individual data points overlaid with the boxplots.

12. Figure 1B: please add numbers in the pie chart (this applies to other pie charts in the manuscript as well).
13. Figure 1C: Many of these properties are correlated eg: mRNA expression levels and protein expression levels. It would be good to show the correlation matrix between properties as a supplementary figure.
14. Figure 1C: The comparison between Essential and nonessential genes should also be shown here. This would provide a better understanding of where exactly dispensable essential genes fall along between essential and nonessential genes in terms of properties.
15. Figure 1E: The average enrichment of absence for core essential genes and nonessential genes sets of the same size should be shown as well. Without these controls, the result is difficult to interpret.
16. Figure 2B, S2H, S2I: No p-values are shown for the enrichments.

We thank the reviewer for the constructive feedback on our manuscript. Below, in red, we have addressed all the individual comments and suggestions. Changes in the manuscript are also indicated in red. We note that the references below include the Pubmed ID, and that we corrected two labels in Figure S2G.

Reviewer #1

In "Meta-analysis of dispensable essential genes: phylogenetic properties and co-occurrence with bypass suppressor mutations across species", Pons and van Leeuwen integrate multiple functional and genomic datasets to explore the properties of previously identified dispensable essential genes. While I have comments on the analysis work that was performed, one of my main concerns with the manuscript has more to do with how the results and methods are presented. As it is currently written, the paper cannot be understood without being intimately familiar with at least 5-6 other manuscripts. In that regard:

1. Please take the time to properly define all terms used in the main text, methods and figure legends. For example, what is the difference between expression variance (g) and expression variance (e) in Figure 1? Some of these properties and where they were obtained are defined only in the methods of van Leeuwen et al, 2020a and b, while for others there is simply no information available.

We included in the Methods a description and a reference/source for each of the 21 features used in the analysis shown in Figure 1C and S3F. To avoid confusion, we removed "expression variance (e)" (Brem & Kruglyak, 2005, 15659551) and kept only "expression variance (g)" (Gasch et al, 2000, 11102521) (now simply "Expression variance"). We extended the Methods to explain in more detail how the feature analysis (Figure 1C and S3F) was performed and added a supplementary table (Table S1) with all the associated statistical data.

Additionally, to increase readability, in Figures 2A, S2D, S2E, and S2G, we added a label indicating the species involved in the corresponding analysis. For clarity, we also modified Figures 4A and S4A.

2. The same is true of genetic interaction screening terminology. Please take the time to explain what you mean by positive or negative genetic interactions and the differences in methodology between the different experiments and datasets reported in the manuscript. While the authors show that the different datasets include genes with similar properties, experimental considerations might also play a role in determining whether the different approaches are expected to yield similar results.

We extended the Results to include a brief definition of genetic interactions and to explain in more detail the observed overlap with genetic interactions. We also added details on the used genetic interaction dataset to both the Methods and the Results, and added a new section in the

Methods to describe the clustering of genetic interaction profiles (Figure 3F). Furthermore, we added information on the differences between dispensable essential gene sets to both the Methods and the Results.

3. Finally, please improve the methods to make them self-supporting. Not every analysis step from previous manuscripts from which datasets originate needs to be re-written, but the rationale and thresholds used should at least be presented so that reading 5-6 different papers is not required to understand what was done here.

*We added three new sections to the Methods: “Overlap with genetic interactions” (Figure 3B), “Clustering of genetic interaction profiles” (Figure 3F) and “On the use of orthology relationships to identify gene activity changes between species” (related to Figure 5 and S5). Besides, we extended the following Methods sections to provide more details about the calculations or the data used: “Dispensable essential gene datasets” (Figure 1A), “Functional enrichment of dispensable essential genes” (Figure S1A), “Properties of dispensable essential genes” (Figure 1C), and “Agreement in copy number changes and suppression mode across *S. cerevisiae* strains” (Figure 4A and S4A). Additionally, we indicate in each Methods section the corresponding figure in the manuscript.*

Regarding the analysis and conclusions presented:

4. The paper examines many properties associated with dispensable genes, which appear to add up to many potential associations being tested. Yet, there is no mention of strategies to control for multiple testing in the methods or the main text. This is further obscured by the fact that the number of associations tested is often not explicit. For example, the authors used the same panel of gene features as in van Leeuwen 2020, but failed to redescribe it in detail here so the total number of properties tested here is unclear. Furthermore, in many cases, associations are only categorized as significant or not, without any test statistics or p-values provided. This is in my opinion one of the most significant weaknesses of the manuscript, as the conclusions of the study fully depend on the robustness of the associations uncovered. Please provide a complete list of associations tested and the strategies used to mitigate multiple hypothesis testing (e.g.: FDR, Bonferonni or other).

We added to the Results and the Methods that the gene feature panel tested in Figure 1C and S3F includes 21 gene and protein-level properties. We listed all the features used and provided the source and/or reference for each of them in the Methods. For numerical and binary features, we evaluated the statistical significance using Mann-Whitney U and Fisher’s exact tests, respectively. To account for the multiple tests performed for each gene set, we calculated the FDR. Comparisons resulting in two-sided p-value < 0.05 and FDR < 10% were considered to be significantly different. This is now explained in detail in the Methods and briefly described in the Results. Additionally, we provided a table with the features tested in Figure 1C and S3F, and the associated p-values and FDR (see Table S1). In all, the enrichments shown in Figure 1C do not change because the comparisons involve large sets of genes (DEGs vs core essential genes,

DEGs vs non-essential genes, and essential vs non-essential genes) which result in very low uncorrected p-values. However, in Figure S3F the compared datasets are smaller (DEGs with multiple suppressors vs DEGs with single suppressors), resulting in weaker uncorrected p-values. After correcting for multiple tests, 2 out of the 5 associations previously found (higher coexpression degree and protein abundance for DEGs with single suppressors) are now not significant.

We also calculated the FDR for Figure S1A and S3B. In S1A, there are minimal changes without consequences to our conclusions. In the S3B top panel, some of the off-diagonal associations (involving DEGs and suppressors of different functional classes) are not significant after using the FDR, reinforcing the fact that DEGs and suppressors tend to be functionally related, our main conclusion from that figure. The S3B bottom panel does not change after using the FDR to correct for multiple testing.

Finally, for every statistical analysis in the manuscript, we added the statistical test used and provided three levels of significance in both the figures and text ($p < 0.05$, $p < 0.005$, and $p < 0.0005$).

5. The level of evidence used to determine if a mutation results in a LOF or GOF in the case of a missense mutation appears very low. Yet, mutations that were classified using SIFT are presented in the text as strictly deleterious or gain of function, without highlighting the fact this is a prediction and providing uncertainty estimates. The accuracy of SIFT can also vary wildly between different proteins, depending on the number of sequences in the input alignment and the protein itself. The authors should provide more information on the confidence associated with their predictions.

We are using the LOF and GOF labels that we and others previously determined experimentally (van Leeuwen et al, 2020, 32939983). In the 2020 study, we tested whether a loss-of-function (deletion or TS mutant) or gain-of-function (overexpression) allele of the suppressor gene could also bypass the need for the corresponding dispensable essential gene. Similar experiments have been performed by studies that are included in the literature dataset (van Leeuwen et al, 2020, 32939983). We have now clarified this in the Results. We are not using SIFT to assign LOF or GOF labels.

*The only link between SIFT and our study is that SIFT was used in Peter et al. 2018 to evaluate the deleteriousness of missense mutations found across *S. cerevisiae* strains. We used that precomputed data (downloaded from <http://1002genomes.u-strasbg.fr/files/>) together with frameshift and indel mutations for the analyses shown in Figures S2A, 4B, S4B, and S4C.*

6. Furthermore, algorithms based on sequence conservation are much better at classifying LOF mutations compared to GOF. There is also nothing preventing a GOF mutation from being non-conservative and thus classified as a LOF by SIFT. The authors fail to mention these

limitations, which have significant implications for several downstream analyses. Factoring in the high potential for false positives in this analysis might for example change the conclusion of the analysis shown in figure 4 and figure 5.

We did not use SIFT to classify mutations as GOF or LOF, as explained in our answer to the previous point. Additionally, we challenged the main results shown in Figure 4 and 5 with extensive controls shown in Figures S4 and S5. It is very unlikely that noise or false positives result in the multiple significant associations that we describe.

7. When discussing CNL and CNG events, the author should explain a bit better how these events were detected in the Peters 2018 dataset. Are only segmental duplications included, or also CNVs due to aneuploidies?

We used the CNL and CNG data as provided by the Peters 2018 study. They used a combination of flow cytometry, sequencing coverage analysis across 1-kb windows, and allele frequency distributions to determine copy number changes, including both aneuploidies and segmental duplications. We have added this information to the Methods.

8. Also regarding CNL and CNG, the authors should explain and support their reasoning as to the effect of duplication on gene activity. Only a small fraction of yeast genes are haploinsufficient, and many of them have relatively flat expression fitness functions (Keren et al 2016, Cell). As the vast majority of natural strains included in Peters et al 2018 are diploids (n=694 diploids out of), shifts to a copy number of 0.5 or 2 might rarely have an effect on fitness, regardless of any potential suppression.

*Copy number changes are associated with variation in gene expression levels and can impact cell growth (Pavelka et al, 2010, 20962780). For instance, even if only ~10% of essential genes were haploinsufficient in rich media conditions (Deutschbauer et al, 2005, 15716499), this increased to ~30% when testing other conditions (Delneri et al, 2008, 18157128), and up to 50% when evaluating high-dimensional phenotypes (Ohnuki and Ohya, 2018, 29768403). Also, expression decrease was deleterious for 47% of the essential genes (Arita et al, 2021, 34096681). Since CNL events of essential genes often result in LOF phenotypes, we screened a panel of *S. cerevisiae* strains for CNL events in DEGs and evaluated if their suppressors presented also a copy number change in agreement with their suppression mode: CNL and copy number gain (CNG) for LOF and GOF suppressors, respectively. However, the consequences of copy number changes in the suppressors are not as clear as for the DEGs. Up to 70% of the suppressor genes are non-essential, and changes in their copy number tend to be less deleterious than for essential genes (Deutschbauer et al, 2005, 15716499). Nevertheless, we previously showed that suppressors do not usually affect fitness on their own (van Leeuwen et al, 2016, 27811238) and only have an effect when combined with the query mutation. We modified the Results to describe the information above.*

9. I strongly disagree that an N:1 orthology relationship is equivalent to a copy number decrease. This hypothesis disregards any possible functional or expression level divergence that could have occurred. The same is true of the opposite, where authors considered 1:N orthology relationships as GOF. The authors should provide convincing evidence to support this reasoning or remove this aspect of the analysis.

A gene duplication after the speciation event generally gives rise to a 1:N orthology relationship. Right at the moment of the duplication event, there is usually an increase in transcript and protein levels (Pavelka et al, 2010, 20962780) causing an increase in gene activity. To evaluate if the initial increase in expression of duplicated genes is maintained in S. pombe, we quantified expression changes between S. cerevisiae (Lipson et al, 2009, 19581875) and S. pombe (Grabherr et al, 2011, 21572440; Koch et al, 2012, 22747640), which are highly correlated ($r=0.62$, $p < 0.0005$, new Figure S5K). For 1:1 orthologs we calculated the S. pombe to S. cerevisiae expression ratio by dividing their expression levels (expression S. pombe / expression S. cerevisiae). For 1:N orthologs (one gene in S. cerevisiae and N in S. pombe), we first added the expression levels in S. pombe of each set of the N orthologs, and then we calculated the same ratio as before. As can be observed in the new Figure S5L, duplicated genes in S. pombe tend to have higher expression ratios than single orthologs (for each transcript in S. cerevisiae, we have more transcripts in S. pombe for duplicated genes than for 1:1 orthologs, medians of 1.345 and 0.791, respectively, $p < 0.0005$ Mann-Whitney U test). Therefore, in general, the expression level increase resulting from a copy number gain is kept in S. pombe. The same logic applies to N:1 orthology relationships and copy number loss (see new Figure S5L).

To account for cases in which the initial copy number changes and the expression changes in S. pombe did not agree, we implemented a new approach to identify genes with increased or decreased activity between S. cerevisiae and S. pombe, by replacing the use of orthology relationships with the quantification of expression changes between both species. For all genes with identified orthologs, we calculated the S. pombe to S. cerevisiae expression ratio between both species as explained above. Genes with an extreme reduction (bottom 5% changes) or increase (top 5% changes) in transcript levels between orthologs were labeled as genes with decreased and increased activity, respectively. By combining expression ratios with the rules based on protein sequences and gene absence (see "Phenotypic changes across species and presence of bypass suppressor mutations"), DEGs with an absent or non-essential ortholog in S. pombe were significantly more likely to have a bypass suppressor in that species (new Figure S5G, $p < 0.005$, Fisher's exact test), in agreement with the result using orthology relationships (see Figure 5B).

However, functional divergence may take place after the duplication event and the initial increase in gene activity may be modulated, which is difficult to evaluate without gene-specific functional assays. To control for the impact to our results of the potential functional divergence after copy number changes, we repeated the same analysis shown in Figure 5B but using only suppressors with 1:1 orthologs in S. pombe or absent in that species. Thus, we did not consider suppressor orthology relationships (i.e. duplicated and N:1 orthologs) or transcript changes to

identify equivalent GOF or LOF bypass suppressors in S. pombe. As can be seen in the new Figure S5F, we still find a significant association between being non-essential or absent in S. pombe and having a bypass suppressor in that species ($p < 0.05$, Fisher's exact test).

We added a new section to the Methods describing these new approaches and the potential problems of using orthology relationships to identify GOF and LOF bypass mutations in S. pombe (see "On the use of orthology relationships to identify gene activity changes between species").

10. The approach used for the identification of suppressor mutations in orthologs completely disregards the potential for cis epistasis that would alter the effect of a potential suppressor mutation. When combined with the probable high false positive rate of the variant effect prediction used by the authors this analysis seems unlikely to be unreliable.

Expression levels between S. cerevisiae and S. pombe highly correlate for 1:1 orthologs ($r=0.62$, $p < 0.0005$, new Figure S5K), which minimizes the risk of suppressor mutations with different effects in S. cerevisiae and S. pombe. Still, the effect of the suppressor mutation in S. pombe could be different in some cases due to additional mutations in the gene, but that would not explain why we observe the multiple significant associations reported in Figure 5 and S5. Additionally, we implemented a new approach in which instead of using orthology relationships to identify gene activity changes similar to GOF and LOF, we used large expression level changes between both species. Therefore, a large expression increase in S. pombe would be equivalent to a GOF suppressor, and a large decrease in expression would be equivalent to a LOF bypass suppressor. Including the expression changes in our methodology resulted in a similar association between DEGs being absent or non-essential in S. pombe and having a bypass suppressor in that species (new Figure S5G).

11. General point for figures: please add the number of data points part of each boxplot, and please consider showing individual data points overlaid with the boxplots.

We added the dataset sizes on the x-axis of the plots and the individual data points on top of the boxplots for figures 2E, 2G, S2B, S2J, S2K, and S2L.

12. Figure 1B: please add numbers in the pie chart (this applies to other pie charts in the manuscript as well).

We added numbers to the pie charts in F1B, S3C, and S3E.

13. Figure 1C: Many of these properties are correlated eg: mRNA expression levels and protein expression levels. It would be good to show the correlation matrix between properties as a supplementary figure.

We added Figure S1F with the Spearman's correlation values between the gene/protein features. Only correlations with $p < 0.05$ and FDR < 10% are shown.

14. Figure 1C: The comparison between Essential and nonessential genes should also be shown here. This would provide a better understanding of where exactly dispensable essential genes fall along between essential and nonessential genes in terms of properties.

We added the comparison between essential and non-essential genes to Figure 1C and removed the previous Figure S1F.

15. Figure 1E: The average enrichment of absence for core essential genes and nonessential genes sets of the same size should be shown as well. Without these controls, the result is difficult to interpret.

We think the reviewer refers to Figure S1E. We added to the panel the corresponding absence fold enrichments for the core essential and non-essential gene sets across the different pangenome sets. We modified the Methods and legend accordingly.

16. Figure 2B, S2H, S2I: No p-values are shown for the enrichments.

We modified the three legends to include the statistical test and p-value threshold used. We also provided Supplementary Table S2 with the p-values obtained in each statistical test. Additionally, for clarity we changed the right plot of each panel which now shows the enrichment for non-essential 1:1 orthologs instead of essential 1:1 orthologs

October 19, 2023

RE: Life Science Alliance Manuscript #LSA-2023-02192-TR

Dr. Carles Pons
Institute for Research in Biomedicine
Baldiri Reixac, 10
Barcelona, Catalonia 08028
Spain

Dear Dr. Pons,

Thank you for submitting your revised manuscript entitled "Meta-analysis of dispensable essential genes and their interactions with bypass suppressors". We would be happy to publish your paper in Life Science Alliance pending final revisions necessary to meet our formatting guidelines.

- please add the Twitter handle of your host institute/organization as well as your own or/and one of the authors in our system
- please add a Category for your manuscript in our system
- please note that titles in the system and on the manuscript file must match
- please upload your main manuscript text as an editable doc file
- please remove figures from the main manuscript text
- please add your main, supplementary figure, and table legends to the main manuscript text after the references section
- please incorporate any points from the Conclusion section into the Discussion; we only allow a Discussion section
- please add an Author Contributions section to your main manuscript text
- please add a Conflict of Interest statement to your main manuscript text
- please provide one figure on a single page
- please add a callout for Figure 5C to your main manuscript text

A. FINAL FILES:

B. MANUSCRIPT ORGANIZATION AND FORMATTING:

Sincerely,

Reviewer #1 (Comments to the Authors (Required)):

The authors have addressed most of my previous comments thoroughly. The additional methodological details have greatly improved the manuscript in terms of readability. While I still disagree with the rationale of certain analyses, there is sufficient information in the method and main text so that readers can make their own opinions on the conclusions.

October 25, 2023

RE: Life Science Alliance Manuscript #LSA-2023-02192-TRR

Dr. Carles Pons
Institute for Research in Biomedicine
Baldiri Reixac, 10
Barcelona, Catalonia 08028
Spain

Dear Dr. Pons,

Thank you for submitting your Resource entitled "Meta-analysis of dispensable essential genes and their interactions with bypass suppressors". It is a pleasure to let you know that your manuscript is now accepted for publication in Life Science Alliance. Congratulations on this interesting work.

DISTRIBUTION OF MATERIALS:

Again, congratulations on a very nice paper. I hope you found the review process to be constructive and are pleased with how the manuscript was handled editorially. We look forward to future exciting submissions from your lab.

Sincerely,
